# Relationships and the Determinants of Sustainable Land Management Technologies in North Gojjam Sub-Basin, Upper Blue Nile, Ethiopia

**Alelgn Ewunetu** [1,2,*] **, Belay Simane** [2] **, Ermias Teferi** [2,3] **and Benjamin F. Zaitchik** [4]

[1] Department of Geography and Environmental Studies, Woldia University, Woldia P.O. Box 400, Ethiopia
[2] Center for Environment and Development Studies, Addis Ababa University,
Addis Ababa P.O. Box 1176, Ethiopia; belay.simane@aau.edu.et (B.S.); ermias.teferi@aau.edu.et (E.T.)
[3] Water and Land Resource Center, Addis Ababa University, Addis Ababa P.O. Box 1176, Ethiopia
[4] Department of Earth and Planetary Sciences, Johns Hopkins University, Baltimore, MD 21218, USA;
zaitchik@jhu.edu
[*] Correspondence: alelgn.ewuntu@aau.edu.et; Tel.: +251-912-772-560

**Abstract:** Sustainable land management (SLM) is a leading policy issue in Ethiopia. However, the adoption and continuous use of SLM technologies remain low. This study investigates the interrelationship of adopted SLM technologies and key factors of farmers' decisions to use SLM technologies in the North Gojjam sub-basin of the Upper Blue Nile. The study was based on the investigation of cross-sectional data obtained from 414 randomly selected rural household heads, focus group discussions, and key informant interviews. Descriptive statistics and Econometric models (i.e., Multivariate Probit and Poisson regression) were used to analyze quantitative data, while a content analysis method was used for qualitative data analysis. Results indicate that at least one type of SLM technology was implemented by 94% of farm households in the North Gojjam sub-basin. The most widely used technologies were chemical fertilizer, soil bund, and animal manure. Most of the adopted SLM technologies complement each other. Farm size, family size, male-headed household, local institutions, perception of soil erosion, livestock size, total income, and extension service increased the adoption probability of most SLM technologies. Plot fragmentation, household age, plot distance, off-farm income, market distance, and perception of good fertile soil discourage the adoption probability of most SLM technologies. To scale up SLM technologies against land degradation, it is important to consider households' demographic characteristics, the capacity of farm households, and plot-level related factors relevant to the specific SLM technologies being promoted.

**Keywords:** sustainable land management; soil erosion; adoption; land degradation; Blue Nile; north Gojjam sub-basin

## 1. Introduction

Land degradation is a major threat to improving rural livelihood strategies in Sub-Saharan African (SSA) countries, where the majority of the population depends on subsistence agriculture [1–4]. In Ethiopia, agriculture is not only the leading economic sector but also a way to build the welfare of society [5,6]. The agriculture sector is the main source of livelihood for more than 80% of the population [7,8], but the productivity of agriculture has been seriously threatened by abject land degradation in the form of soil erosion and soil nutrient depletion [5,6,9–13]. The country has struggled to feed its growing population, and the problem may become more severe in the near future if agriculture yields are not increased [9]. The problem is particularly critical in the highland regions, where the majority of the population depends on crop–livestock mixed subsistence agriculture systems [9,14]. The major causes of land degradation in the Ethiopian highlands are rapid population growth, climate variability, top-down planning systems, poor implementation of policies, limited use of SLM technologies, and frequent organizational restructuring [9,15].

Recognizing the need to slow and reverse land degradation in order to achieve food security, the Ethiopian government has prioritized the adoption of SLM technologies since the 1970s [14,16]. Over the past decades, various SLM development projects and strategies have been designed and employed by the government and through foreign initiatives [17,18]. Nevertheless, adoption rates and sustained use of SLM techniques have been below expectation [14,16]. The focus on physical soil and water conservation (SWC) technologies, inappropriate installation of SLM technologies, top-down implementation approaches, and a lack of farmers' motivation to invest in land management in the long run are among the reasons for these failures [16,17,19]. Partly as a result, the country continuously loses a tremendous amount of topsoil resources [19–27]. For example, it is estimated that about 33.7 t ha$^{-1}$yr$^{-1}$ topsoil has been lost from the northwestern Ethiopian highlands [20]. Similarly, on average, about 27.5 t ha$^{-1}$yr$^{-1}$ topsoil has been lost from the entire Upper Blue Nile basin [25]. Soil erosion has been an acute problem in the North Gojjam sub-basin of the Upper Blue Nile; about 45.3% of the sub-basin was found to experience high to very high soil loss risk, with an average soil loss of 46 t ha$^{-1}$yr$^{-1}$ [19].

Bearing in mind these constraints, the government of Ethiopia has gradually shifted to new SLM strategies, approaches, and technologies for smallholder farmers using extension agents and technical experts to realize access to infrastructure, training, sustainable energy resources, and agricultural technology inputs [28–30]. Since 1995 the government has integrated several SLM technologies into agricultural extension packages through community mass mobilization at the watershed level [17,18]. These include the SLM Project (SLMP), Making Environmental Resource to Enable Transition to More Sustainable Livelihood (MERET) project, and Public Works Programme of the Productive Safety net Programme (PSNP) [30]. Agricultural intensification was also pursued, including the implementation of external agricultural inputs such as improved seed, chemical fertilizer, and agronomic technologies [31–33]. Despite the benefits of SLM technologies, the adoption rate still remains low, and degradation continues to constitute a fundamental challenge for productivity [19,25]. Thus, identifying the current constraints of farmers' decision to adopt SLM technologies is important to inform revised policies and strategies, recognizing that local land users' social, financial, human, and physical assets endowment and capacity are dynamic [2,4,10,13].

Several studies have addressed factors affecting the household's decision on SLM technologies adoption in the Ethiopian highlands and elsewhere. However, previous studies have focused on a single technology and ignored complementarity and trade-offs between SLM practices [16,26,34,35]. Additionally, the majority of these studies modeled the implementation of SLM technologies as a binary variable (adopter and non-adopter) and did not account for the intensity of SLM adoption. These types of modeling approaches cannot capture the preference of land users' behavior for different SLM technologies and simultaneous acceptance or non-acceptance decisions [36–39]. Since SLM technology adoption choices are multivariate, it is necessary to consider simultaneous and consecutive decision-making processes and potential trade-offs related to the adopted SLM technologies. Nigussie et al. [40] address determinants of simultaneous SLM technologies, but that study lacked access to several important variables, such as compost and improved seed adoption factors. To address this weakness in the literature, this study aims to analyze the relationship and determinants of sustainable land management technologies in the North Gojjam sub-basin of the upper Blue Nile. This is accomplished through the following specific objectives: (1) To analyze major determinants that influence farmers' decision to adopt SLM technologies; (2) To identify factors that affect farmers' choice to adopt a set of SLM technologies; (3) To show the interdependency of past adopted SLM technologies.

The analyses provided in this study offer a better understanding of relationships and determinants of farmers' adoption behavior and are, therefore, important for designing enabling environments to stimulate the adoption of SLM technologies in a sustainable manner and ultimately bring agricultural productivity. The farmers' decisions related to the adoption of SLM technology in the upstream part of the Upper Blue Nile will have

an effect on the lifespan of water reservoirs and irrigation canals located downstream. These decisions are also relevant for strategies to reduce siltation of the Ethiopian Grand Renaissance Dam (GERD) because the North Gojjam sub-basin is a primary source of water for the dam. The remainder of the paper is organized as follows: Section 2 presents an overview of the study area, the survey design and data, specification of econometric models used for estimation, concepts of SLM technologies, and independent variables used in this study. Section 3 provides the analysis results and implications for SLM decisions. In Section 4, the conclusions are drawn, and further suggestions are noted.

## 2. Research Methodology

### 2.1. Description of the Study Area

The North Gojjam sub-basin is located between 38.2° E and 39.6° E longitude and 10.8° N and 11.9° N latitude. The sub-basin is found in Amhara regional state, mainly on the border of East Gojjam, West Gojjam, and South Gondar Zones (Figure 1). The sub-basin is one of the major headstreams of the Blue Nile (Abbay) River and covers 1,431,360 ha. The total population of the sub-basin and the surrounding villages was 3,565,892 in the most recent census [41]. The population settlement is dispersed in the sub-basin, on average ranged from 260 to 270 people per km$^2$, and the majority is living in foot small mountain areas by forming small villages with relatives. The average population density in the sub-basin was 204.17/km$^2$, with lower 117 to higher 288/km$^2$ [41]. The mountainous areas and the upland are less populated relative to the middle land. The altitude of the sub-basin ranges from 1044 to 4048 above mean sea level (amsl). The dominant agro-ecological zone is characterized by tepid to cool moist middle highlands and cold to very cold moist sub-afro-alpine to afro-alpine highlands, but the eastern and southeastern parts of the sub-basin are hot to warm moist lowlands. The dominant climate condition of the sub-basin is the tropical highland monsoon [42]. According to the Ethiopian National Meteorological Agency (EMA) [43], the average maximum and minimum temperature of the sub-basin varies from 24.6 °C to 28.1 °C and 11.0° to 14.5 °C, respectively, and the mean annual temperature is 19.4 °C. The rainfall pattern is closely correlated with the annual migration of the Inter-Tropical Convergence Zone (ITCZ), and most of the rainfall occurs in summer, from June to September, with a maximum in August [44]. The distribution of rainfall across the sub-basin is uneven; the highland tends to be wetter than the lowlands. The meteorological record data from the stations within and the surrounding areas of the sub-basin (1986–2017 years) indicate that the mean annual rainfall is 1334.5 mm with a minimum of 810 mm and a maximum of 1815 mm [43].

The dominant soil types are leptosols, vertisols, luvisols, and alisols. The geology of the sub-basin is mainly dominated by basalt, but the lowland is dominated by sandstone [45]. Land uses include: cultivated land cover 70.67% of the sub-basin, grazing land (12.49%), bush and shrubland (8.21%), forest cover (2.07%), tree plantation (3.61%), and bare land (2.91%; Figure 2). Natural forest cover is very low and is found primarily on riverbanks, hillsides, and churches. Natural forests are particularly concentrated around churches because the surrounding community is afraid to cut trees on the church's property [27].

The *eucalyptus globules* forest is the dominant introduced tree, particularly in plantations in the highland regions, and its cover has increased through time. Unreliable rain-fed agriculture is the primary source of livelihood for the majority of the population in the sub-basin. It is characterized by smallholder mixed crop and livestock production systems. The dominant crop types are cereals, pulses, and oilseeds [27]. Dominant livestock types are cattle, sheep, goat, horse, donkey, mule, and poultry. Soil erosion, climate variability, and water shortage explain to a large extent the prevailing food insecurity and poverty in the sub-basin.

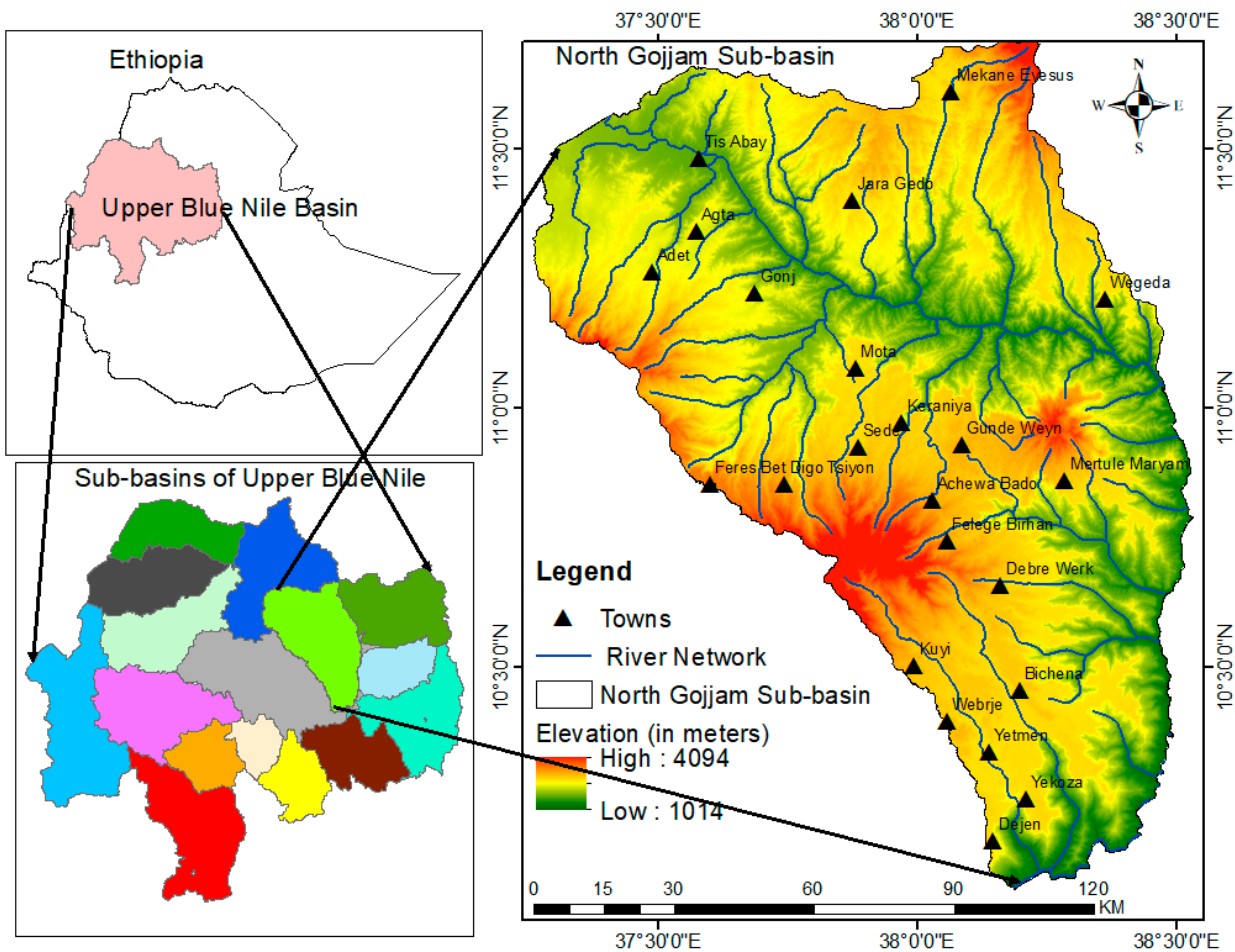

**Figure 1.** Location of the North Gojjam sub-basin.

### 2.2. Sample Size and Sampling Techniques

A multistage sampling method was used to acquire the data for this study. Based on the information obtained from a panel discussion with stakeholders and the authors' prior knowledge, the North Gojjam sub-basin was selected from the Upper Blue Nile sub-basins. Based on population density, administration, and geographical location criteria, three districts were selected from the sub-basin (Table 1). Then, nine rural villages (three from each district) were selected from the upper, middle, and lower parts of the district (Table 1). Finally, 414 household heads were selected from the lists of farm households acquired from the respective local development agents' offices. The sample size in the respective districts and villages was determined using the proportional stratifying sampling method (Table 1).

**Table 1.** Distribution of sample households by agro-ecology and administration.

| Zone | District | Village | Agroecology | Total HH | Sample HH |
|---|---|---|---|---|---|
| East Gojjam | Enarj Enauga | Koso-zira | Highland | 932 | 34 |
| | | Titar Badima Yizar | Middle land | 1151 | 43 |
| | | Gedeb Georgis | Low land | 1649 | 61 |
| West Gojjam | Dega Damot | Ziqual Wogem | Highland | 1154 | 43 |
| | | Arefa Medhanyalem | Middle land | 1120 | 42 |
| | | Gense-Tekeleaaymanot | Low land | 642 | 24 |
| South Gondar | Andabet | Gota | Highland | 1644 | 61 |
| | | Yedidi Gimegne | Middle land | 1250 | 46 |
| | | Genete Mariyam | Low land | 1616 | 60 |
| Total | - | - | - | 11,158 | 414 |

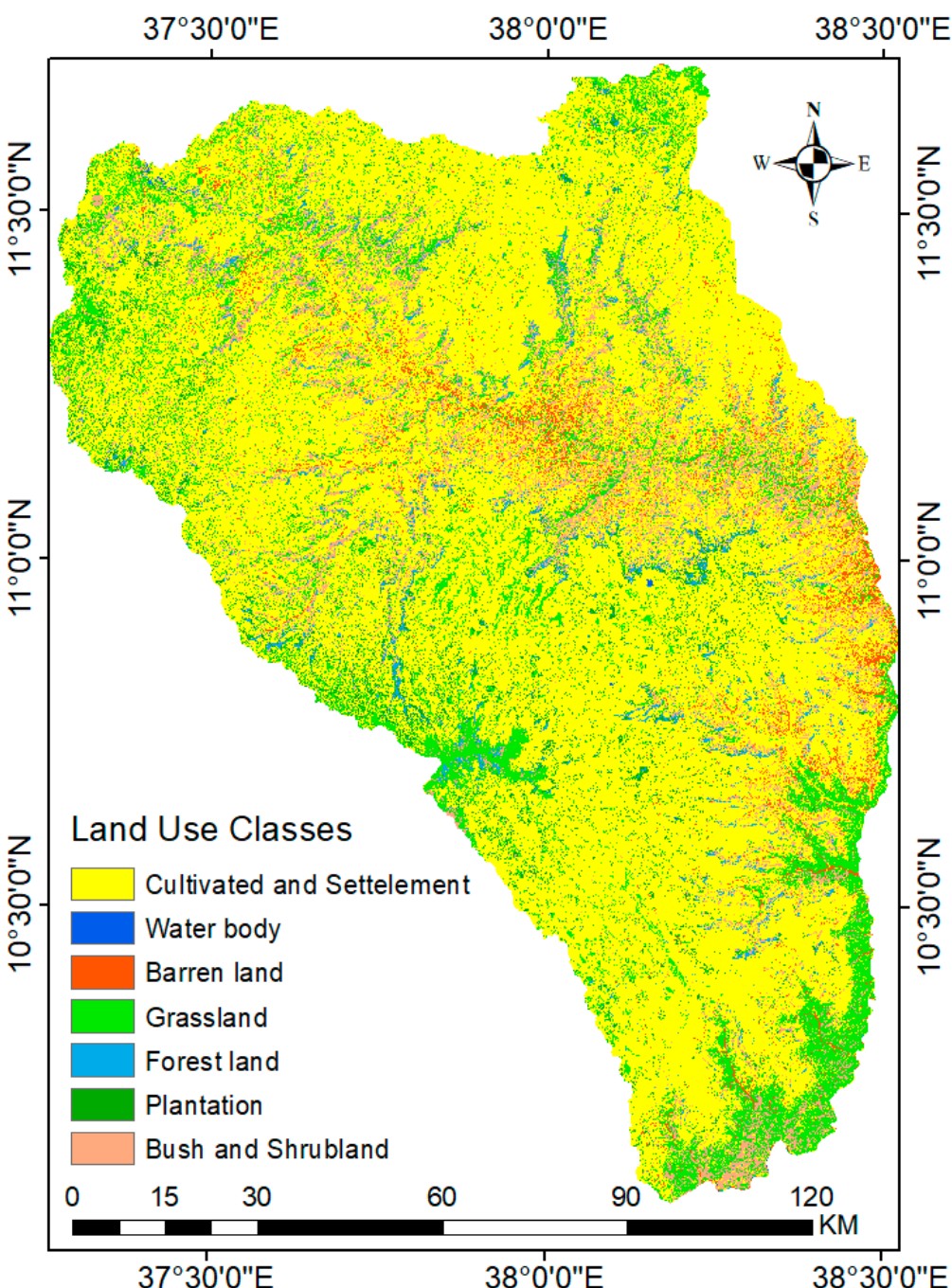

**Figure 2.** 2017 land use map of the North Gojjam sub-basin.

In addition, 127 Focus Group Discussion (FGD) participants were selected from nine villages and different social classes based on their age, gender, and local knowledge. Key informants were selected from farmers, Development Agents (DAs), and districts' agricultural and natural resources experts.

*2.3. Data Sources and Data Collection Methods*

The cross-sectional data for this study were collected using both open and close-ended questionnaires in May and June 2018. The questionnaire had different demographic, topographic, plot, infrastructure, and institutional characteristics. In addition, it included land users' perceptions on the status of soil erosion and soil fertility status. The reliability and validity of the questionnaire were established based on: (1) experience in earlier

studies; (2) input from local development agents; (3) pre-testing on 20 randomly selected farm households in a similar district. Based on the feedback obtained, some questions were omitted, amended, refined, and rearranged. Finally, it was translated to the local language (Amharic). The questionnaire was administered by well-trained enumerators under the close supervision of the first author. In addition, a series of FGDs and in-depth interviews were conducted to complement and contextualize the quantitative data. In each FGD meeting, participants were limited to 6–10 members. Each FGD meeting was facilitated by the first author guided by a checklist. In each study village, two series of FGDs were held, and in-depth interviews were employed for different communities.

*2.4. Method of Data Analysis*

Field data were analyzed using descriptive statistics and econometric models. Descriptive statistics were used to describe households' socioeconomic characteristics, land users' perception of soil erosion and fertility status, and SLM technology adoption status. A multivariate probit model was employed to analyze the interrelation of adopted SLM technologies and the determinants of farmers' decision to adopt SLM technologies, and a Poisson Regression Model was employed to analyze the intensity of adopted SLM technologies (analysis performed in STATA14). Qualitative data obtained from the key informants and FGDs participants were analyzed using a content analysis to complement the quantitative results.

Econometric Framework and Estimation Strategies

Farm households always consider technology in the context of multiple objectives and options. Selected SLM technologies may be used simultaneously or sequentially to solve a range of farmland problems [37–39]. Because the adoption of SLM technology is inherently multivariate, the use of the binary probit model is inefficient because it ignores the correlation in the error terms of adoption equations [39,46]. Failure to capture this interdependence leads to unfair conclusions [37,46]. Therefore, to analyze the causal relationships and determinants of farmers' SLM technology adoption, we deploy a multivariate probit (MVP) model following the procedures of Nigussie et al. [40]. In this study, the MVP comprises seven binary choice equations that address soil bund, stone bund, compost, manure, improved seed [1], agroforestry, and inorganic fertilizer. Hence, we specified the study model as:

$$Y_{im}^* = \beta_m X_{im} + \varepsilon_{im} (m = 1, 2 \ldots 7) \tag{1}$$

$$Y_{im} \ \{1 \ if \ Y_{im}^* > 0 \ and \ 0 \ otherwise$$

Equation (1) is developed based on the assumption that a rational $i^{th}$ farm household has a latent variable $Y_{im}^*$ that captures unobserved factors related to the $m^{th}$ choice of SLM technologies (m = 7 SLM technologies). $X_{im}$ comprises exogenous variables that determine SLM adoption, such as households' socioeconomic, demographic, institutional, and plot characteristics (Table 2). The coefficients $\beta_m$ quantify the effects of the vector of dependent variables. $\varepsilon_{im}$ represent error terms following a multivariate normal distribution, each with a mean of zero and a variance–covariance matrix with values of 1 on diagonal and non-zero correlations as off-diagonal elements [47].

A Poisson Regression Model (PRM) was used to analyze the determinants of farmers' decisions to adopt SLM technologies in the context of multiple adoption options for each plot and decision time. Fo llowing [49], the PRM model of the dependent variable $(y_i)$, which was constructed as the sum of the binary responses of the SLM measures implemented on the plot by the $i^{th}$ rational farmer, is specified as:

$$E (y_i) = \beta x_i + \varepsilon_i \tag{2}$$

where $E (y_i)$ is the expected value of the dependent variable for the $i^{th}$ rational farmer, $\beta$ is the set of parameters that reflect the impact of change, $x_i$ is a vector of observed variables, and $\varepsilon_i$ refers to the error terms in the model result.

**Table 2.** Descriptive and summary statistics of the variables used in this analysis.

| Variable's Name | Variable Description (Coding/Units) | Expected Sign | Mean | Standard Deviation |
|---|---|---|---|---|
| Gender | Household head gender type, 1 = Male, Female = 0 | ± | 0.82 | 0.38 |
| Age | Farm household head's age (years) | − | 50.32 | 14.83 |
| Education | Educational status of household head, (years) | + | 1.56 | 0.68 |
| Family size | Number of family members (count) | ± | 5.38 | 2.19 |
| Dependency ratio | The ratio of members aged below 15 and above 64 to those aged between 15 and 64 (count) | − | 0.81 | 0.73 |
| Farm size | Area of cultivated land, (hectare) | ± | 1.03 | 0.71 |
| Plot number | Land fragmentation (count) | − | 2.95 | 1.81 |
| Farm distance | Plot distance to the residence (minutes of walking) | − | 29.31 | 17.82 |
| Steep slope | The slope of a farmland perceived as a very steep = 1 | ± | 0.21 | 0.37 |
| Moderate slope | The slope of a farmland perceived as a moderate = 1 | ± | 0.43 | 0.49 |
| Gentle Slope | The slope of a farmland perceived as a gentle = 1 | + | 0.36 | 0.48 |
| Soil fertility | Farmland soil status perceived as good fertility = 1 | ± | 0.35 | 0.47 |
| Soil erosion | Farmland perceived as high soil erosion = 1 | ± | 0.54 | 0.49 |
| Market distance | Market distance to the residence (minutes of walking) | − | 122.04 | 57.69 |
| Training | Household received SLM-related training = 1 | + | 0.39 | 0.48 |
| Extension adv. | Household received SLM-related advice = 1 | + | 0.62 | 0.49 |
| Media | Access to newspapers, own radio/TV/mobile = 1 | + | 0.35 | 0.47 |
| Membership | Participation in village clubs = 1 | + | 0.87 | 0.34 |
| Access to credit | Household received credit = 1 | + | 0.34 | 0.48 |
| TLU | Livestock herd size (tropical livestock unit; TLU) | ± | 4.13 | 2.66 |
| Income | Household total annual income per annum (ETB) | ± | 67,664.64 | 14,556.47 |

Note: ± indicates a mixed result expectation. ETB is the Ethiopian Birr (currency). For TLU calculation, Calf = 0.25, Donkey (young) = 0.35, Weaned calf = 0.34, Heifer = 0.75, Goat/sheep (adult) = 0.13, Cow and ox = 1.0, Goats/sheep (young) = 0.06, Horse = 1.10, Donkey (adult) = 0.70, Chicken = 0.013 [48].

*2.5. Concepts of Sustainable Land Management (SLM) Technologies*

In the context of this study, SLM refers to knowledge-based integrated resource management to achieve food security and sustainable ecosystem services. SLM technologies can be grouped as physical, agronomic, and vegetative management measures implemented to reduce soil erosion and soil nutrient depletion and enhance soil productivity [1,18]. Farmers use a range of SLM technologies in the Ethiopian highlands [17,18]. In this study, we consider soil and stone bunds to be an example of a structural intervention used to reduce erosion, compost, and manure applications. An example of an agronomic intervention promoted by extension agents in the study area is using chemical fertilizer applications at the planting time. Another agronomic intervention, one that is popular among farmers, and improved seed and agroforestry, is a vegetative intervention that provides multiple benefits through the joint management of trees (e.g., for animal forage) and cereal crops [10,50].

*2.6. Explanatory Variables Considered in This Study*

The choice of explanatory variables in this study was made based on economic theory and the empirical literature on determinants of SLM and agriculture technology adoption [14,16,17,35–37,51–53]. These include socioeconomic factors (i.e., age, gender, education level, family size, dependent ratio), plot-specific attributes (i.e., farmland size, plot distance to the residence, slope position of the farmland, land users' perception of soil erosion, and soil fertility status), SLM related training and advice, off-farm income, access to credit, livestock size, and total households' income. The descriptions of the designated variables hypothesized the direction of influence, and descriptive statistical measures are presented in Table 2.

## 3. Results and Discussion

### 3.1. Characteristics of the Respondents

Table 2 presents the description and the mean values of variables used in the models. The majority of the respondents were male-headed (83%), while the minority (17%) were female-headed, who were mainly divorced or widows. The average age of the study household heads was 50 years, and the minimum and maximum ages were 22 and 85 years old, respectively. The survey result indicates that about 46.5% of respondents were illiterate, while 53.5% can read and write. Their family size range was from 2 to 11 members and had 5.4 average family sizes, and the average dependency ratio was 0.8.

The surveyed household had a 1.03 ha average landholding size, which ranged from 0.15 to 4 ha, and their plots were often not spatially adjacent; thus, they cultivated on average three plots. In the study area, plots take on average a 58 min round-trip walk from the farmer's residence. Most plots were used for annual cereal–legume crop production. While evaluating farmland characteristics, on average, about 21% of respondents perceived that their farmland was steep, 43% moderate, and 36% of the land had a plain slope. About 35% of respondents perceived that their cultivated land was fertile. The majority of farmlands were perceived by the land users to have moderate to very severe erosion status.

The respondents were also asked to reply regarding contractive advice from the DAs. Sixty-two percent of respondents reported that they had received advice, while 39% of respondents attended formal SLM-related training. Thirty-five percent of respondents had their own radio/mobile, and 87% of the sampled households were involved in a farmers' cooperative group. Farmers in the study area walk about two hours on average to arrive at the nearest agricultural input–output market from their house. They possessed 4.13 tropical livestock units (TLU) on average. Eighteen percent of respondents were engaged in off-farm activities, and 45% of sample households had received credit from formal institutions. Moreover, the descriptive statistics presented in Table 2 illustrated that respondents had an average annual income of ETB 67,664.64 (USD1 $\approx$ ETB 29) during the survey season.

### 3.1.1. The State of SLM Technology Adoption in the North Gojjam Sub-Basin

Farmers in the North Gojjam sub-basin used various SLM technologies. Based on government and extension agent priority, seven SLM technologies were considered in this study, including soil bund, stone bund, manure, compost, inorganic fertilizer, improved seed, and agroforestry (Figure 3). Among the total respondents, 69.1% used soil bund, while 44.4% used stone bund. The spacing and depth of both soil and stone bunds construction vary from plot to plot depending on the slope gradient and topsoil characteristics of the farmland. The result of the FGD and in-depth interview confirmed that few farmers deliberately destroyed structural SLM technologies because they are thought to provide shelter for rodents, present barriers to plowing, and are a waste of fertile cropland areas. Some farmers also perceived that these structures had a poor performance for controlling soil erosion.

Chemical/inorganic fertilizer was used by 74.4% of the respondents in the North Gojjam sub-basin. Most farmers used less than the recommended amount of chemical fertilizer due to a lack of finance to purchase and the inaccessibility of the supply. According to in-depth interviews, the use of manure was limited around the homesteads due to the shortage of manure. Of the total respondents, about 59% used manure in the study area on at least one plot. Similarly, compost application was used by a small number of respondents (32%) during the survey season due to lack of awareness, shortage of inputs, and the labor required to prepare it. The improved seed was used by 57% of the respondents during the survey season on at least one plot of land. The interviewees argued that the improved seed is not accessible at the right time and right place to be used in the study area. The use of agroforestry practice on private farmland appears to remain low in the North Gojjam sub-basin. It was practiced by about 32% of respondents during the survey year. The majority of local farmers were not well-aware of agroforestry technology adoption and its benefit. Generally, the result shows that structural and agroforestry SLM practices

were more used in the Dega Damot district, whereas inorganic fertilizer was more widely implemented in the Andabet district. Conversely, the improved seed was adopted more in the Enarj Enauga district. Of the seven SLM technologies, inorganic fertilizer was most widely used, followed by soil bund and manure in the sub-basin during the survey season.

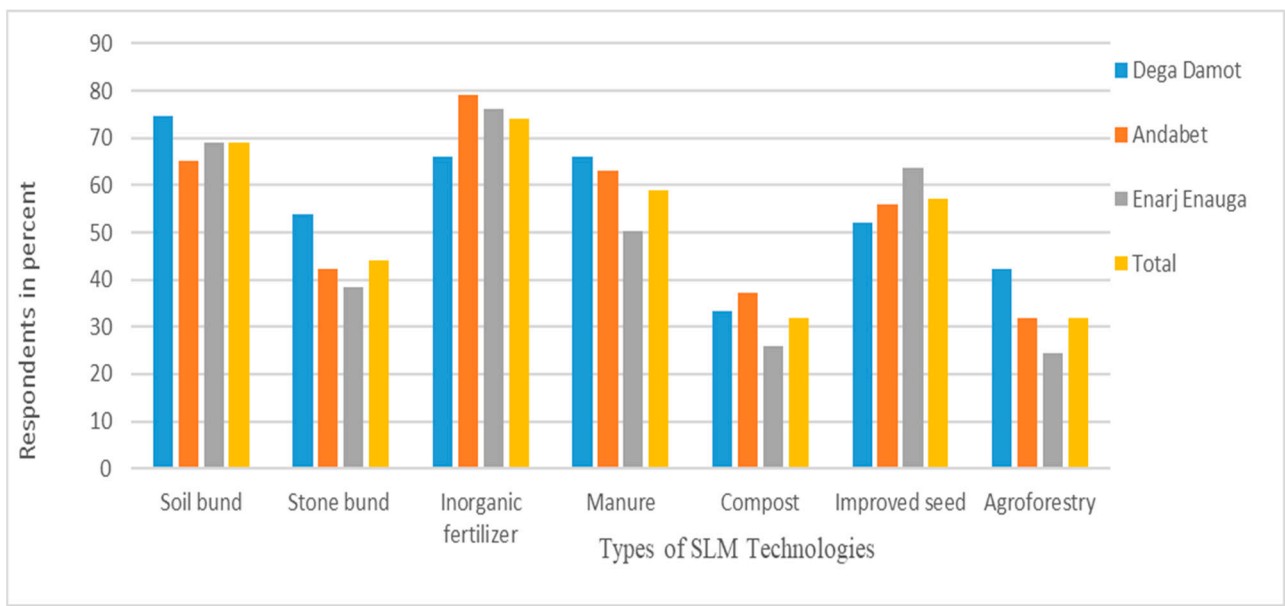

**Figure 3.** Types of SLM technologies used across districts in the North Gojjam sub-basin.

The few SLM technology adoption differences among the districts may have resulted from administrative, agro-ecology, and cropping pattern variations. The Dega Damot district is located in the west Gojjam zone, and its dominant agro-ecology is the *dega*/temperate zone, and the stable crop is potato. The Andabet district is located in the south Gondar zone, and its dominant agro-ecology is the *Woina dega*/sub temperate, and wheat is the dominant crop. On the other hand, the Enarj Enauga district is located in the east Gojjam zone, and its dominant agro-ecology is *Woina dega*/sub temperate zone. Teff is the dominant crop type in this district.

### 3.1.2. Number of Adopted SLM Technologies in the North Gojjam Sub-Basin

Among the seven SLM technologies, land users used from 0 to 7 SLM technologies simultaneously on their plot during the survey season (Figure 4). The result shows that about 6% of the respondents have not applied any of the studied SLM technologies, though those households sometimes used other traditional SLM measures, such as ditches. The use of multiple SLM measures was prevalent: the plurality of households used two SLM in the survey season, and many used more. This simultaneous use of multiple SLM practices is indicative of the interdependence between SLM adaptation decisions. The result shows the intensity of the SLM adoption rate across the three districts was almost similar.

### 3.2. *Empirical Results and Discussion*

Tables 3 and 4 show the MVP and PR model analysis results. The results of the PR model fit the data well (p $\cong$ 0.00; log-likelihood ratio ($X^2$ (22) = 60.48); Pseudo R$^2$ = 0.0685). Similarly, the choice of the MVP model is justified by the significance likelihood ratio test ($X^2$ (21) = 108.76, p = 0.00).

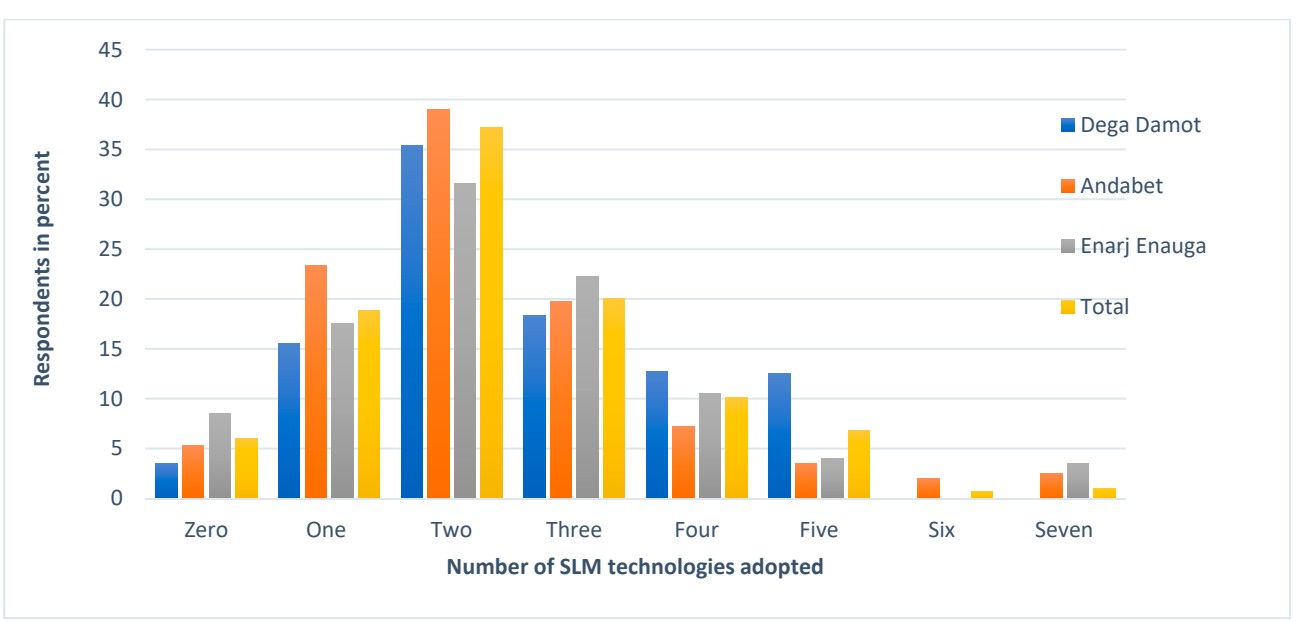

**Figure 4.** Number of SLM technologies adopted in the North Gojjam sub-basin.

**Table 3.** MVP joint covariance matrix of regression results between adopted SLM technologies.

|  | $\rho^{SO}$ | $\rho^{ST}$ | $\rho^{MA}$ | $\rho^{CO}$ | $\rho^{FE}$ | $\rho^{IS}$ | $\rho^{AG}$ |
|---|---|---|---|---|---|---|---|
| $\rho^{SO}$ | 1 |  |  |  |  |  |  |
| $\rho^{ST}$ | 0.029 (0.116) | 1 |  |  |  |  |  |
| $\rho^{MA}$ | −0.347 (0.122) *** | −0.067 (0.109) | 1 |  |  |  |  |
| $\rho^{CO}$ | −0.091 (0.125) | −0.115 (0.105) | 0.682(0.087) ** | 1 |  |  |  |
| $\rho^{FE}$ | 0.573 (0.119) *** | 0.195 (0.136) | −0.109 (0.129) ** | −0.264(0.136) ** | 1 |  |  |
| $\rho^{IS}$ | 0.407 (0.120) *** | 0.157 (0.099) | 0.118(0.102) | 0.005(0.106) | 0.766(0.072) *** | 1 |  |
| $\rho^{AG}$ | 0.296 (0.119) ** | 0.081(0.105) | −0.077(0.108) | −0.029(0.109) | 0.065(0.133) | 0.109(0.109) | 1 |

rho Likelihood ratio test: $\rho SOST = \rho SOMA = \rho SOCO = \rho SOFE = \rho SOIS = \rho SOAG = \rho STMA = \rho STCO = \rho STIS = \rho STAG = \rho MACO = \rho MAFE = \rho MAIS = \rho MAAG = \rho COFE = \rho COIS = \rho COAG = \rho FEIS = \rho FAIS = \rho FAAG = \rho ISAG = 0$: chi2(21) = 108.76, Prob > chi2 = 0.00 **, *** significant at 5%, and 1% levels, respectively. SO = soil bund, ST = stone bund, MA = manure, CO = compost, FE = chemical fertilizer, IS = improved seed, and AG = agroforestry. In parenthesis are standard errors.

### 3.2.1. The Relationship Among the Adopted SLM Technologies

The regression results in Table 3 indicate the presence of interdependence between the seven adopted SLM technologies'. Among the positive and significant correlations, the highest (0.8) was observed between improved seed and inorganic fertilizer. Soil bund and chemical fertilizer (0.6); improved seed and soil bund (0.4); soil bund and agroforestry (0.3); and compost and manure (0.7) are positively and significantly correlated to each other. Given the different timescales associated with implementing different technologies, these findings imply that the adoption of physical land management technology induces the application of short-run land management technologies. Soil bund promotes the application of agroforestry such as animal fodder and various plantation trees. Adoption of improved seed highly encouraged the use of chemical fertilizer. Moreover, farmers in the sub-basin used compost and manure jointly. This is consistent with the study conducted by Teklewold et al. [38], which applied an MVP model and found a positive association between improved seed and inorganic fertilizer adoption is the highest (42%), but that manure can be a substitute for inorganic fertilizer in rural Ethiopia.

**Table 4.** MVP and Poisson Regression (PR) results of single and number of SLM technology adoption determinants.

| Variables | Multivariate Probit (MVP) Model | | | | | | | PR Model |
| | Soil Bund | Sone Bund | Manure | Compost | Fertilizer | Imp. Seed | Ag/Forest | No. SLM |
|---|---|---|---|---|---|---|---|---|
| Age | −0.043 (0.094) | 0.005 (0.007) | −0.021 (0.008) *** | −0.033 (0.008) *** | 0.016 (0.010) ** | −0.004 (0.006) | 0.002 (0.01) ** | −0.004 (0.003) |
| Gender | 0.218 (0.289) * | 0.761 (0.319) ** | −1.04 (0.328) *** | 0.608 (0.318) * | 0.523 (0.330) | 0.419 (0.264) | 0.255 (0.313) | 0.547 (185) *** |
| Education | 0.116 (0.053) ** | −0.081 (0.129) | 0.103 (0.136) | −0.180 (0.143) | 0.893 (0.215) ** | 0.204 (.125) | −0.016 (0.132) | 0.009 (0.063) |
| Household size | 0.762 (0.222) *** | −006 (0.042) | 0.132 (0.050) *** | 0.0292 (0.053) * | 0.127 (0.059) ** | 0.052 (0.041) | 0.020 (0.045) | 0.083 (0.038) ** |
| Farm size | 0.014 (0.006) ** | 0.325 (0.162) ** | 0.039 (0.045) | −0.091 (053) ** | 0.061 (057) * | 0.115 (0.164) | 0.920 (0.221) *** | 0.036 (0.02) *** |
| Plot Number | −0.173 (0.078) ** | −0.036 (0.057) | −0.105 (0.06) * | −0.160 (0.073) ** | 0.146 (0.248) | 0.028 (0.057) | −0.156 (0.067) | −0.001 (0.029) * |
| Plot distance | 0.014 (0.006) ** | −0.000 (0.005) | −0.011 (0.005) * | −0.617 (0.258) ** | −0.002 (0.005) | −0.010 (0.004) * | −0.015 (0.005) *** | −0.004 (0.002) |
| Plain slope | 0.543 (0.342) | −0.629 (0.290) ** | 0.543 (0.261) ** | 0.558 (0.301) ** | 0.278 (0.383) | 0.304 (0.242) | −0.463 (0.261) * | −0.012 (0.117) |
| Moderate Slop | 0.677 (0.287) ** | 0.116 (0.186) | 0.157 (0.194) | 0.267 (0.291) | 0.155 (0.302) | 0.267 (0.182) | 0.025 (0.200) | −0.036 (0.095) |
| Soil fertility | 0.138 (0.273) | 0.046 (0.247) | −0.015 (0.217) ** | 0.088 (0.228) | −0.362 (0.348) | −507 (0.224) * | −0.233 (0.197) | −0.026 (0.103) |
| Soil erosion | 0.947 (0.251) *** | 0.173 (0.212) ** | −0.064 (0.208) | −0.018 (0.211) ** | −0.173 (0.279) | 0.360 (0.216) * | 0.393 (0.221) ** | −0.026 (0.103) |
| Market distance | 0.000 (0.001) | 0.001 (0.000) | −0.0004 (0.0007) | 0.0001 (0.000) | 0.0017 (0.001) | −0.031 (0.014) ** | −0.0005 (0.0006) ** | −0.128 (0.040) *** |
| Training | 0.767 (0.257) *** | 0.045 (0.183) | 0.376 (0.197) * | 0.688 (0.198) *** | 0.726 (0.262) *** | 0.449 (0.177) *** | 0.169 (0.191) | 0.206 (0.094) *** |
| Expert advise | 0.096 (0.249) | 0.129 (0.186) | −0.249 (0.199) | 0.093 (0.200) * | 0.020 (0.236) | −0.091 (0.163) ** | 0.461 (0.205) ** | −0.070 (0.095) |
| Media | 0.506 (0.349) | −0.343 (0.262) | 0.023 (0.279) * | 0.205 (0.302) | 0.418 (0.32) ** | 0.033 (0.231) * | −0.329 (0.278) | −0.046 (0.157) |
| Membership | 0.375 (0.330) | −0.318 (0.262) | −0.270 (0.284) | 0.213 (0.319) | 0.313 (0.294) ** | 0.033 (0.234) ** | −0.411 (0.275) | −0.046 (0.130) |
| Credit | 0.088 (0.236) | −0.153 (0.175) | 0.159 (0.188) | 0.088 (0.191) | 0.933 (0.292) *** | 0.011 (0.162) ** | −0.197 (0.187) | −0.074 (0.090) |
| Livestock | −0.017 (0.046) | 0.049 (0.036) | 0.342 (0.051) *** | 0.094 (0.412) ** | 0.124 (0.048) ** | 0.029 (0.036) | 0.008 (0.038) | 0.013 (0.017) |
| Off farm | −0.001 (0.000) *** | −0.000 (0.000) | $−3.60 \times 10^{-6}$ (0.000) | −0.00001 (0.00001) | 974 (0.377) ** | 0.0415 (0.028) * | −0.000 (0.000) | −0.000 (0.000) |
| Income | $−2.60 \times 10^{-6}$ (0.000) | $−6.50 \times 10^{-6}$ ($9.80 \times 10^{-6}$) | 0.00014 ($9.98 \times 10^{-6}$) ** | $9.80 \times 10^{-7}$ (0.000) | 0.419 (0.184) ** | 0.0002 (000) ** | 0.00002 ($9.80 \times 10^{-6}$) ** | $9.20 \times 10^{-6}$ ($4.80 \times 10^{-6}$) * |
| Constant | −1.77 (0.837) ** | −1.57 (0.680) ** | 1.04 (0.692) | 0.960 (0.704) | 0.750 (0.875) ** | −0.420 (0.629) | 0.133 (0.680) | 0.457 (0.361) |
| Wald chi$^2$ | (140) = 313.0 | | | | | | | $X^2(21) = 108.8$ p = 0.00 |
| Prob > chi$^2$ | 0.000 | | | | | | | |

\*, \*\*, \*\*\* significant at 10%, 5%, and 1% levels, respectively. Number of observations = 258, Standard errors in parentheses, No. = Number.

In contrast, the use of some technologies was found to be negatively associated. Manure and inorganic fertilizer (0.6), and compost and inorganic fertilizer (0.2) were correlated negatively and significantly. This implies that the two external agricultural inputs, inorganic and organic fertilizers, substitute for one another or have different availabilities for different household types. The application of organic fertilizer is labor-intensive, whereas chemical fertilizer is capital-intensive. Hence, households with financial resources but limited labor are likely to purchase chemical fertilizers, while those that have sufficient labor or insufficient finance may opt for manure and compost. The findings of this study are in line with the previous empirical studies of Teklewold et al., Mengistu and Assefa, and Sileshi et al. [38,54,55], who reported that farmers face various land degradation problems, and thus, they used more than one SLM technology.

### 3.2.2. Determinants of Farmers' Decision to Adopt of SLM Technologies

The MVP and PR model results in Table 4 reveal that household age has mixed effects on SLM technology adoption decisions. Age is positively associated with the adoption of inorganic fertilizer and agroforestry at a 5% significant level. However, it is negatively associated with the probability of animal manure and compost adoption at a 1 and 5% significant level, respectively. This shows that older farmers are less likely to adopt labor-intensive SLM technologies, and it might be explained by the fact that younger farmers might be more physically strong to collect, prepare, and transport the bulk materials to their farmland. In contrast, older farmers may have more assets to invest in chemical fertilizer. This finding is similar to those of other studies of Asfaw and Neka, Teklewold et al., and Saguye [35,38,53], which reported that the age of the household head was negatively and significantly correlated with manure application. Using a binary probit model, Miheretu and Yimer [34] also confirmed that older farmers often invest more in inorganic fertilizer because they have more assets than younger ones. Similarly, older farmers are more likely to adopt agroforestry because they have a larger farm size. The implication is that if farmers have small land sizes, they are reluctant to use their farmland for agroforestry. The result is in line with the finding of Etsay et al. [8], who found that age positively influenced the adoption of agroforestry because younger farmers have small farm sizes in that study area.

The gender of the household head has mixed results on the adoption probability of different SLM technologies (Table 4). A male-headed household is more likely to adopt stone bund at 5% and soil bund, compost, and a set of SLM practices at a 1% significant level. Parallel to this, a study conducted by Asfaw and Neka [35] also found that male-headed households are more likely to use structural SLM practices. The possible reasons are that male-headed households are better exposed to SLM technology information and have more labor and assets to adopt than female-headed ones, in part because female heads of the household are more likely to be burdened with non-agricultural work such as childcare and food preparation. In the North Gojjam sub-basin, female farmers tend to depend on the help of male farmers near them or relatives elsewhere to carry out their agricultural activities. However, female-headed households are more likely to implement animal manure at a 1% significant level than males. The evidence from qualitative information also shows that in the study area, manure-related work is traditionally a female duty, which offers a possible explanation for this pattern. This manure adoption result is consistent with the findings of Belay and Bewket [56] from a study in northwestern Ethiopia.

Education is known to play an important role in enhancing technology adoption decisions [57]. We find a significant association between higher levels of household head education and the adoption of soil bunds and chemical fertilizer at a 1% significant level, respectively. The finding of a positive association between farmers' educational level and these SLM technology applications is consistent with previous studies [35,37,53,55] that have identified that educational status has a positive effect on the farmers' decision to retain introduced SLM technology, though we did not find a significant association for all SLM technologies or for adopting sets of SLM practices.



Family size had a positive effect on the adoption of soil bund, manure, compost at 1%, and chemical fertilizer, and a number of SLM technologies at a 5% significant level (Table 4). This result is in line with the findings of Gebremedhin and Swinton and Miheretu and Yimer [14,34], who stated that the presence of more household members' favors adopting labor-demanding SLM technologies. The same view was also stressed by FGDs and key informant interview participants in the North Gojjam sub-basin. These informants explained that there are no other options to produce food for the family in this locality other than subsistence agriculture. Thus, the family labor force often engages in on-farm activities to produce more food. However, Amsalu and de Graaff and Sileshi et al. [16,55] found contradictory results and explained that larger families might be less likely to adopt SLM practices because there is a competition for labor between food generating off-farm activities and SLM work. Likewise, Shiferaw, and Holden [58] specified that structural SLM measures occupied a wide area and occupied the scarce productive land resource. Therefore, farmers with large family sizes may be inclined to remove the constructed SLM technologies from their farm fields.

The relationship between farm size and the adoption of SLM technologies could, in principle, go in either direction. Larger farms are indicative of greater assets and reduced risk aversion, and hence a higher probability of adopting SLM practices. At the same time, small farm size might push for intensification and allow for greater investment of labor and inputs per unit area [53,56]. A household that works a larger extent of farmland needs high labor and time to build conservation structures on this larger land area, as well as to keep up the fertility status of their plots. In this study, we found a mixed result from farm size on different SLM technology adoption decisions. Farm size is positively correlated with the adoption of soil bund and stone bund at 5%, agroforestry at 1%, and inorganic fertilizer and a number of SLM technologies at a 10% significant level. However, large land size decreases the application of compost at a 10% significant level (Table 4). The result shows that structural SLM practices and agroforestry occupied land space, and for small farms, the profit from conservation may not be enough to substitute for the decline in production due to the loss in the area used for conservation structures. Hence, farmers who have small land sizes are less likely to adopt such SLM types. Instead, they use technologies that do not occupy the area (i.e., manure and compost). Similar findings were reported from other regions in Ethiopia by Amsalu and de Graaff, Mengistu and Assefa, and Ahmed et al. [16,54,59], which all found that structural land management technology positively correlated with farm size. However, Asfaw and Neka [35] found a negative association between farmland size and the probability of adopting structural SLM practices. They attribute this to the fact that most farm households that have large farm sizes are old-aged, and they thus have short-term plans and lack the labor force required to construct structural SLM measures.

The likelihood of farmers to adopt all soil bund, manure, compost, and multiple SLM technologies declined at a 1% significant level with an increase in plot fragmentation (Table 4). This could be because land that is dispersed and fragmented usually creates a wastage of time and labor, as time is required to reach each plot. In addition, land users explain that fragmented farmland leads to challenges when installing structural SLM measures and agroforestry because these technologies can lead to disagreements with the adjacent farm owners unless they are willing to use similar SLM technologies. This finding is similar to the result of Asfaw and Neka, Sileshi et al., and Bekele and Drake [35,55,60], who confirmed that a large number of fragmented lands discourages the adoption probability of SLM technologies.

As may be expected, the choice of SLM technologies differs as a function of farmland slope. Farmland with gentle slopes was more likely to be treated with both manure and compost at a 5% significant level. Land with gentle slopes was less likely to receive stone bund conservation structures at a 1% significant level (Table 4), while moderately sloped lands are significantly more likely to be the target of soil bund measures. Qualitative focus group and key informant results confirm that stone bunds are valued as an erosion-

reducing intervention on steep slopes. The application of manure and compost is very low in such steep terrain because the land is more vulnerable to soil erosion associated with high-speed runoff. To prevent such erosion problems, land users choose to implement stone bunds, cutoff drains, and traditional ditches. This general result is consistent with other studies [14,34,55,59].

It is expected that a stronger land user perception of land degradation would be associated with greater adoption of SLM practices [53]. Consistent with this expectation, we found that the adoption probability of increased soil bund at 1% and stone bund and agroforestry at a 5% level of significance with farmer perception of the severity of erosion, which is similar to findings in other studies [53,55]. A higher perception of erosion was negatively associated with the probability of manure application at a 1% significant level, perhaps because of the expectation that manure would be washed away. Interestingly, farmer perception of soil fertility was negatively associated with adoption probability of animal manure and improved seed at a 5% significant level. This may be because farmers who perceive that their plots are fertile direct resources away from those high-performing plots.

Farm distance influences the choice of farmers' SLM technology adoption in different ways. The present study confirmed that plot distance negatively affected the use of compost at 5% and both manure and agroforestry at a 1% significant level, whereas it positively associated with the use of soil bund at a 1% significant level (Table 4). The information from the qualitative analysis confirms that distance plots present challenges for transporting compost and manure. Similarly, the adoption of agroforestry in the farthest plot is difficult due to free grazing. This negative association found here is consistent with Teklewold et al. [38]. The positive association for soil bunds is inconsistent with the findings of Asfaw and Neka [35], who reported that distance from home to cultivated land is negatively associated with the adoption of the probability of structural SWC measures.

There is an expectation that proximity to the market could encourage investment in SLM, given the higher potential to sell the product at a favorable price and general accessibility of inputs. Nevertheless, the evidence on the impact of market accessibility on SLM technology adoption in Ethiopia is unclear [37]. We find that proximity to the market is positively associated with the use of chemical fertilizer, improved seed, and agroforestry at a 1% significant level but negatively associated with the number of SLM technologies applied. The finding of a negative association between market distance and most of the leading SLM is consistent with the studies of Saguye [53] and Teklewold et al. [36], which stated that distance to the market has a negative and significant effect on the adoption of land management technologies, reflecting transaction and access cost.

Training for farmers is a vital institutional factor that increases awareness about the benefits of SLM technology adoption. As indicated in Table 4, access to SLM-related training increases the adoption probability of chemical fertilizer at 5% and all soil bund, manure, compost, agroforestry, and the total number of SLM technologies adopted at a 1% significant level. However, access to training was found to be rare in the study area. Of the total respondents, about 39% have attended formal SLM-related training in the survey season (Table 2). This suggests that SLM-related training and smallholder farmers are the key factors for smallholder farmers to inspire the adoption of integrated SLM practices. This result is in line with the findings of several previous studies in Ethiopia: Zeweld et al., Mengistu and Assefa, Saguye [2,53,54], which reported that access to training has a positive and significant effect on the adoption of introduced SWC technologies. However, Bekele and Drake [60] found that access to SLM relating training had no significant impact on the probability of new land management technology adoption.

Contact with development agents (DAs) is another vital institutional factor that can increase the adoption probability of SLM technologies. We analyzed the relationship between farmers' satisfaction from experts' advice and the adoption probability of SLM technologies. The results show that advice from the DAs increases the probability of compost, improved seed at 1%, and agroforestry adoption at a 5% significant level (Table 4).

Similarly, Asfaw and Neka, and Bekele and Drake [35,60] reported that farm households who gain better information from extension agents are willing to invest in new SLM technologies and are more likely to sustain them. Conversely, Zeweld et al. [2] found that extension service had a negligible and even negative impact on the adoption of SLM technologies.

We found that households who have access to mass media (radio/TV) are more likely to adopt manure and improved seed at 1% and chemical fertilizer at a 5% significant level (Table 4). The result implies that access to information through mass media increases the use of manure and purchased farm inputs but has an insignificant effect on the use of structural SLM measures and agroforestry. The reason seems to be that agricultural inputs are promoted in local broadcasts. On the other hand, farmers who are a member of traditional working cooperatives can acquire information about agricultural technologies independent of messages received from mass media or extension services. In line with this, the result of this study confirmed that membership in a farm cooperative is significantly and positively associated with the use of improved seed at 1% and both stone bunds and chemical fertilizer at a 5% significant level (Table 4). Similarly, farmers in the open interview agreed that the cooperative's social network provided access to shared labor, finance, and knowledge relevant to implementing SLM technologies. This suggests that to upscale the introduction of SLM practices, local institutions and service providers may want to encourage and assist farmers' cooperatives at a local level.

In this region, livestock ownership is an indicator of wealth. The livestock are also fundamental components of farming systems in the study region, including their role in buffering against crop failure and income shocks. In the context of SLM, manure and compost accessibility depend on the size of the herd a household owns [53]. As we hypothesized, livestock size was positively associated with the adoption probability of manure at 1% and both compost and chemical fertilizer at a 5% significant level (Table 4). Correspondingly, qualitative information reported that livestock contribute to households' general wealth, serves as a source of draft power, manure, and transportation, and as a source of cash income to purchase external farm inputs such as chemical fertilizer and improved seed. The result is consistent with findings of Kassie et al., Teklewold et al., Saguye, and Belay and Bewket [4,38,53,56], who found that livestock had a significant and positive influence on the use of organic fertilizer.

Access to credit services has a mixed influence on the various adoption of SLM technologies. Regarding this, we found that access to formal credit increases the use of purchased agricultural inputs such as chemical fertilizer and improved seed at a 5% significant level (Table 4). This is consistent with the finding of Miheretu and Yimer and Holden et al. [34,61]. In contrast, the potential impacts of off-farm opportunities on several SLM technologies are ambiguous [15]. Off-farm activity may enable households to access additional income to hire extra labor and purchase external agricultural inputs, but such opportunities may also discourage on-farm activities. Regarding this, the MVP analysis (Table 4) shows that off-farm income has a significant negative impact on the adoption of soil and stone bunds at a 5% significant level. However, off-farm income increased the probability of inorganic fertilizer and improved seed adoption at a 1% significant level. That is, a household head's access to off-farm income discourages the use of labor-intensive technologies and encourages the use of purchased technologies. Previous studies have consistently shown a negative relation between off-farm opportunity and the adoption of structural SLM technologies [16,35]. Our result showed a mixed result on different SLM technologies and implied the relationship between off-farm activities and SLM adoption decision depends on the type of SLM technology.

Not surprisingly, we found that farm households with a higher income were more likely to adopt animal manure, compost, chemical fertilizer, improved seed at 5%, and a set of SLM technologies at a 1% significant level (Table 4) than those with low income. The result of the present study is similar to the finding of Kassie et al. and Sileshi et al. [4,55], who reported that farmers with high incomes are more be able to take risks, including

those associated with testing new SLM technologies. Wealthier households are also simply more be able to pay for purchased inputs.

## 4. Conclusions and Policy Implications

Land degradation is a grave environmental problem, as it leads to low agricultural productivity and persistent poverty in the Ethiopian highlands. Understanding the determinants of a household's decision to adopt various land management technologies is critical to prevent land degradation and improve rural livelihoods. The main objectives of this study were to investigate the relationships between the adoption of various SLM technologies and to identify determinants of a household's decision to adopt one or multiple SLM technologies. Both quantitative and qualitative data were collected from communities in the North Gojjam sub-basin of the upper Blue Nile basin. Descriptive statistics and econometric models were applied to analyze the quantitative data, while the content analysis method was applied to quantitative data.

The finding shows that 94.2% of respondents reported implementing at least one type of SLM technology on one plot of land. The most widely used SLM measures were inorganic fertilizer (65%), stone bunds (53.8%), and animal manure (50.5%). The result of the MVP regression analysis revealed both complementarities and substitutability among the studied SLM technologies. The highest complementarity was observed between improved seed and chemical fertilizer. This implies that the adoption of the improved seed depends on the supply of chemical fertilizer and vice versa. The adoption of purchasing farm inputs (i.e., chemical fertilizer and improved seed) depends on structural SLM technologies, mainly on soil bunds. A strong complementary was also observed between the use of soil bunds and the adoption of agroforestry techniques. Substitutability, meanwhile, was found for soil fertility inputs: farmers tended to apply either chemical fertilizer or organic fertilizer.

Turning to the drivers of SLM practices, factors positively associated with increased likelihood to implement at least one SLM practice include farm size, family size, male household head, participation in local institutions, perception of soil erosion, number of livestock, household income, and presence of extension service. Plot fragmentation, household age, plot distance, off-farm income, market distance, and farmers' perception of good soil fertility, meanwhile, were negatively associated with the adoption of most SLM practices. The probability of a household adopting multiple SLM measures was higher for larger family size, larger plot size, participation in training, and household income. Market distance and land fragmentation were associated with decreased likelihood to adopt multiple SLM technologies.

Based on these findings, we draw the following policy recommendations for sustainable development in the developing countries in general and in the North Gojjam sub-basin in particular. First, the portfolio of SLM technologies available to farmers exhibits both complementarities and substitutability. These relationships between SLM should be considered when promoting any specific SLM to farmers. Second, a household's decision to adopt SLM technologies is directly related to demographics, socioeconomic conditions, and plot characteristics. Accordingly, these factors should be standard considerations for extension efforts and SLM-oriented development strategies for specific technologies and for holistic land management initiatives. Third, a specific finding is that neighboring landowners need to work together on SLM in order to reduce the negative impact that land fragmentation has on SLM adoption. Fourth, a number of factors generally understood to be important for promoting SLM were confirmed to be relevant in the North Gojjam sub-basin: environmental education, access to media, access to SLM-related training and education, financial support for SLM activities, regular extension communication, and timeliness of availability of agricultural inputs. Fifth, the integration of livestock-cropping systems is critical in this region and in many other subsistence agriculture contexts. This is particularly clear in this study when it comes to the ability to adopt manure and com-

post technologies. Thus, livestock productivity and management opportunities should be considered when promoting cropland SLM practices.

In closing, we recognize the limitations of this study and the need for future research. One priority area is to identify the determinants of farmers' decision to discontinue using improved energy-saving stove technologies, as most households in the sub-basin ignored improved stove technologies introduced by extension agents. While stoves were not a primary focus of this study, relationships between the stove fuel source and biomass management and labor are relevant for SLM practices. Another important topic is to identify challenges of afforestation and reforestation survival and sustainability after planting, as the tree planting efforts in the area have had limited success due to tree death. Finally, more research is required to identify locally appropriate perennial crops and multipurpose trees for agroforestry in order to improve the uptake and productivity of agroforestry techniques in the study region.

**Author Contributions:** Conceptualization, A.E. and B.S.; methodology, A.E., B.S. and E.T.; software, A.E.; validation, A.E.; B.S. and B.F.Z.; formal analysis, A.E.; investigation, A.E.; B.S. and B.F.Z.; resources, A.E.; data curation, B.S. and E.T.; writing—original draft preparation, A.E.; writing—review and editing, B.F.Z.; visualization, E.T.; supervision, B.S. and E.T.; project administration, B.S.; funding acquisition, B.F.Z. All authors have read and agreed to the published version of the manuscript.

**Funding:** This research was funded by NILE-NEXUS: Opportunities for a sustainable food–energy–water future in the Blue Nile Mountains of Ethiopia. Belmont Forum. Addis Ababa University and John Hopkins University have also supported this research.

**Institutional Review Board Statement:** Informed consent was obtained from all subjects involved in the study.

**Informed Consent Statement:** Not applicable.

**Data Availability Statement:** The datasets analyzed in this study are available from the corresponding author on reasonable request.

**Conflicts of Interest:** The authors declare no conflict of interest.

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
