# Peer review of "Relationships and the Determinants of Sustainable Land Management Technologies in North Gojjam Sub-Basin, Upper Blue Nile, Ethiopia"

_sustainability, doi:10.3390/su13116365_

Round 1

Reviewer 1 Report

The paper has the character of a local study with weaker theoretical and methodological  contribution. It is very well processed. The problem is that it has only character a local study. It can be a valuable contribution to the national level,  for environmental policy-making and decision-making processes. The theoretical and methodological impact is weaker, because it applies common sociological and statistical methods.

In the introduction, it would be necessary to specify in more detail the aim and benefits of the paper.  Lines 76 - 79 state the objectives of the work. What is Goal 3? Is it just wrong numbering? It would also be desirable to provide a more detailed description of the baseline situation and to highlight what new contribution will the study bring.

Information about elements of land use could be included in the description of the area of interest for better understanding of the situation. It would be very useful to include a land use map too.

The abbreviations used in the tables should be explained below the tables.

The methods are suitable for processing the issue. The results are described in  detail and very well, but the discussion   missing. The discussion must be completed and connected.  In the discussion it would be necessary to compare the results and the approaches with the results of other authors. It would be desirable to evaluate what new from a theoretical-methodological aspect  this paper brings.

The conclusion need to be expanded.  It would  be necessary to specify in more detail the research questions and problems that need to be further investigated. It would be appropriate to indicate how the results could be used in developing policies. The findings of the study can provide valuable support in developing policies aimed at implementation sustainable development technologies.  Is it possible to draw conclusions and recommendations for other countries on the basis of a local study?

Author Response

Please find attached herewith our revised manuscript, entitled “The Relationships and the Determinants of Sustainable Land Management Technologies in North Gojjam sub-Basin, Upper Blue Nile”, which has been revised to address the comments of the reviewer. We addressed all comments and issues raised by the reviewer, and we feel that the manuscript has definitely improved through this revision.

For a detailed response on the comments, please see below Table.  All authors would be happy to answer any further question, and we are looking forward to the results of your final assessment of the manuscript.

Reviewer 2 Report

The article deals with determinants of sustainable land management in Ethiopia. It is quite a straightforwardly written article, with a clearly presented methodology and interesting results. Although English is not my native language, however, as to my understanding the text strongly requires a review by a proficient English speaker.

I would recommend the article for publication in the journal after extensive English proofreading, and after the following issues will have been clarified:

Lines 8-9: Who has the 4th affiliation? Apparently, you missed putting it.

Line 18: Please, capitalize North Gojjam.

Line 30: “…Sub-Saharan African (SSA)…”. Did you miss the word “countries”?

Lines 30-32: It seems that the second phrase repeats the idea from the first phrase. Please, consider changing it by avoiding such repetitions.

Line 34: Did you mean “… the development of the agricultural sector…”?

Lines 76-79: It seems that you missed objective no. 3.

Line 91 and other similar citations all across the text: If you use the full name in Figure names, then you should cite them in full (Figure X).

Line 94: I can guess what does ‘masl’ means, however, you should avoid such abbreviations, or, at least, give the names in full on their first use.

Figure 1: Taking into consideration the high difference in altitude, from 1 to 4 thou. m, adding a physical-geographic map would be highly helpful to understand better the setting.

Line 112: You write “Natural forest cover is very low and found on river banks, hillsides and churches”. Why churches are included here, along with river banks and hillsides? I believe this interesting and strange idea should be explained.

Table 1: You have the same description for two variables: “Plot distance to residence”. Once it refers to the Farm distance variable, the second time it describes the Market distance one. It is misguiding. You should clarify it.

Line 232: I believe you should explain how tropical livestock units are calculated because not all the journal’s readers are aware of these units.

Lines 338-339: You write “Research conducted by … consists of our finding…”. It is not clear. Please, consider re-phrasing it.

Line 494: You write “… inorganic fertilizer and vis-à-vis.” It is not clear. Please, consider re-phrasing it.

Author Response

(The authors gave the same response as above.)

Reviewer 3 Report

Dear authors,

After analyzing this manuscript, I believe this is an interesting candidate for a thematic journal as LAND; however, once it focuses on sustainability and provides us some policy implications about this typology of development, it also falls within this journal scope.

Generally, the work is satisfactory. It is a well-structured paper, defines a research problem, and finally gives us some policy implications.

However, in my opinion, for this manuscript, proceed for further editorial processing stages; the following issues should be revised/improved:

  • some figures do not present good quality (i.e., of figure 2); improve
  • a scheme about the used methodology could help understand the steps taken by the authors
  • a section regarding the study limitations and prospective research lines should be added
  • please ensure this work follows the MDPI template

Best regards

Author Response

(The authors gave the same response as above.)

Round 2

Reviewer 2 Report

After improvements, the article looks much better. I would recommend it for publication in present form.

Author Response

Thank you.